# Isolation of Carboxylic Acids and NaOH from Kraft Black Liquor with a Membrane-Based Process Sequence

**DOI:** 10.3390/membranes13010092

**Published:** 2023-01-10

**Authors:** Silvia Maitz, Lukas Wernsperger, Marlene Kienberger

**Affiliations:** Institute of Chemical Engineering and Environmental Technology, Graz University of Technology, Inffeldgasse 25c, 8010 Graz, Austria

**Keywords:** black liquor, carboxylic acids, lignin, bipolar membrane electrodialysis, biorefinery, nanofiltration

## Abstract

In kraft pulping, large quantities of biomass degradation products dissolved in the black liquor are incinerated for power generation and chemical recovery. The black liquor is, however, a promising feedstock for carboxylic acids and lignin. Efficient fractionation of black liquor can be used to isolate these compounds and recycle the pulping chemicals. The present work discusses the fractionation of industrial black liquor by a sequence of nanofiltration and bipolar membrane electrodialysis units. Nanofiltration led to retention of the majority of lignin in the retentate and to a significant concentration increase in low-molecular-weight carboxylic acids, such as formic, acetic, glycolic and lactic acids, in the permeate. Subsequent treatment with bipolar membrane electrodialysis showed the potential for simultaneous recovery of acids in the acid compartment and the pulping chemical NaOH in the base compartment. The residual lignin was completely retained by the used membranes. Diffusion of acids to the base compartment and the low current density, however, limited the yield of acids and the current efficiency. In experiments with a black liquor model solution under optimized conditions, NaOH and acid recoveries of 68–72% were achieved.

## 1. Introduction

Recent developments regarding the price and supply of fossil resources, global climate change and greenhouse gas emissions underline the need to change from a fossil-based economy to a system based on renewable resources. The pulping industry can play an important role within this transition process as it processes vast quantities of biomass all through the year. Nowadays, the predominant chemical pulping process is the kraft process [1]. During kraft pulping, around 50% of the wood is dissolved in the black liquor (BL) [2]. Besides the dissolved wood degradation products, BL also contains sodium and sulphides originating from the inorganic pulping chemicals NaOH and Na_2_S. The organic fraction in BL consists mainly of lignin, carboxylic acids, minor fractions of carbohydrate fragments and extractives [3,4]. After kraft cooking, the weak black liquor with a dry matter content of 15% is concentrated by evaporation of water before it is burned in the recovery boiler to regenerate the pulping chemicals and to produce steam and power for the pulp and papermaking processes [5]. Instead of their use as a fuel in the recovery boiler, part of the BL’s organic fraction can be recovered prior to combustion. At some mills, the dissolved kraft lignin is already separated from BL by acid precipitation. This versatile aromatic compound can then be used, e.g., as an external fuel or as replacement for fossil-based compounds. An example is the phenol replacement in the production of phenol/formaldehyde resins [6,7,8]. Besides lignin, carboxylic acids can also be isolated from the BL prior to combustion. This is of special interest as their heating value is low compared to that of lignin [9], and they are essential building blocks for a wide range of products, e.g., pharmaceuticals [10], food additives [11] and polymers [12]. However, isolation of carboxylic acids from BL has so far only been realized at the lab scale. Here, the focus typically lies on the economically interesting low-molecular-mass acids such as formic acid, acetic acid, glycolic acid and lactic acid [2]. Complex sequences of separation technologies are applied, amongst which are nanofiltration (NF), chromatography, ion exchange, distillation, crystallization and esterification [13,14,15,16,17,18]. Electrodialysis (ED), a membrane process, as a treatment method for BL has so far mainly been tested to separate lignin or sodium hydroxide [19,20]. Kumar and Alén also applied ED to regenerate sodium hydroxide and sulphuric acid from spent pulping liquor, but they also reported to have directly isolated carboxylic acids with ED. They stated that a carboxylic acid yield of 80% was achieved based on ED experiments with carbonated, dilute soda-anthraquinone liquor [14]. Another process sequence applying ultrafiltration coupled with two different ED treatments resulted in a patent, claiming the isolation of low-molecular-mass aliphatic carboxylic acids from BL [21].

ED can be applied to provide H^+^ and OH^−^ ions without the need to add or discharge any salts, acids or bases. It can also be used to separate cations and anions in water desalination or to upgrade aqueous effluents [22]. Similar to NF, ED is already largely applied on an industrial scale in water and wastewater treatment [23,24] as well as for the purification of valuable compounds such as carboxylic acids [22] and in the food industry [25]. In general, ED equipment consists of a stack of ion selective membranes between a cathode and an anode that provide an electric field. For treatment of BL streams, two different types of ED can be considered, conventional electrodialysis (CED) and electrodialysis with bipolar membranes (EDBM). In CED, cation-exchange membranes (CEM), which only cations can pass through, and anion-exchange membranes (AEM), which only anions can pass through, are arranged alternately in the electric field. In EDBM, ion-impermeable bipolar membranes (BPM) that essentially consist of a combination of cation- and anion-exchange membranes are additionally used. In an electrical field, the solvent molecules in the aqueous layer inside a BPM can be split into H^+^ and OH^−^ ions. Different feed streams are cycled between the membranes to achieve the desired separation of ions.

With an EDBM membrane arrangement consisting of alternately stacked CEMs, AEMs and BPMs, a BL feed containing carboxylates, sulphides, lignin and sodium ions can be treated to isolate and protonate the dissolved carboxylic acids and at the same time generate a purified stream of NaOH that can be recycled to the pulping process. The present work aims to describe and investigate such a process in detail to give a solid basis for further studies on the application of ED for the treatment of BL and to highlight the opportunities and the challenges associated with such a process. The feed pretreatment by NF, the electrodialytic migration of carboxylic acids and lignin, the recovery of inorganics as well as current efficiencies and energy requirements are investigated based on a BL feed stream and a BL model solution.

## 2. Materials and Methods

### 2.1. Materials

Calibration of the high-performance liquid chromatography (HPLC) system for carboxylic acid quantification was carried out with formic acid (98–100%) from Chem-Lab (Zedelgem, Belgium) and acetic acid, glycolic acid and sodium lactate (all >99%) from Sigma-Aldrich (St. Louis, MO, USA). The acid model solution for the ED experiment was prepared with formic acid (98–100%) from Chem-Lab, L-lactic acid (90%) from Roth (Karlsruhe, Germany) and glycolic acid (99%) and acetic acid (99.8%) from Sigma-Aldrich. Sodium hydroxide pellets (98%) for pH adjustment were purchased from Thermo-Fisher Scientific (Waltham, MA, USA).

The pretreated kraft BL was supplied by a local pulp mill. Twelve different wood species were processed by the mill, amongst which were beech, birch, acacia and oak. The liquor composition and sample characteristics are summarized in Table 1. Besides the components listed in Table 1, BL also contains inorganic pulping chemicals and salts such as sodium, sulphur compounds and carbonate as well as organic compounds such as other carboxylic acids, hemicelluloses and extractives that were not quantified.

### 2.2. BL Feed Pretreatment by NF

To minimize the risk of fouling of the ED membranes, the feed BL was subjected to NF to reduce the concentration of dissolved lignin. For that, the feed was diluted 2:3 with deionized water and then treated in an OSMO Memcell system (OSMO Membrane Systems, Korntal-Münchingen, Germany), which comprises a single-membrane module with a flat sheet membrane of 80 cm^2^ membrane area. The process was operated in batch mode; the retentate was recycled to the feed tank while the permeate was gathered in the product compartment. The feed temperature was 60 °C and the trans-membrane pressure was 32–34 bar, the feed flow rate to the membrane module was approx. 4.3 L/h. Fractionation was performed in several batches with polymeric flat sheet membranes from Microdyn NADIR and Filmtec (now Dow Chemical Company, Midland, MI, USA), NP010P, with a molecular weight cut-off of 1000–1200 Da. Typically, the feed was treated until a volume reduction of 35–50% was reached. Several batches of NF were conducted to yield a sufficient amount of permeate for the EDBM experiment.

### 2.3. EDBM Experiments

The laboratory-scale ED set-up, shown in Appendix A, was purchased from Hescon GmbH (Engstingen, Germany). It consists of a membrane stack in EDBM configuration, four compartments with separate circulation loops and the power unit EA-PS 3032-10 B laboratory power supply (EA, Viersen, Germany). The experiments were performed in batch mode; the feed solutions were cycled through the stack and then back into the respective compartment. The flow rate through the membranes was monitored with flow meters and pressure gauges and controlled to assure a constant pressure drop across the three feed compartments for the acid, the base and the salt feed. This resulted in liquid flow rates of 60–80 L/h. The pressure of the fourth compartment with the electrode rinse (electrolyte) solution was not monitored. The temperature throughout the experiments was not controlled and increased over the course of 1 h until it reached a constant value at 34 ± 1 °C. The feed volume in each compartment was 2 L. The membrane stack was arranged in EDBM mode (see Figure 1), fumasep FKB-PK-130 cation exchange membranes, fumasep FAB-PK-130 anion exchange membranes and fumasep FBM-PK-130 bipolar membranes were purchased from Fumatech BWT GmbH (Bietigheim-Bissingen, Germany). The membranes were separated by polymeric spacers that provided the flow and direction of the feed streams through the stack and protected the membranes from mechanical damage. The membrane stack consisted of five cell triplets, [CEM-AEM-BPM]_5_. The active membrane area of each individual membrane was 100 cm^2^. Two different types of EDBM experiments were conducted with this equipment and membrane configuration: one with pretreated BL permeate from the NF process, the other with a model solution.

#### 2.3.1. EDBM Treatment of BL Permeate

The feed of the salt compartment was the pooled permeate of the NF experiments with pretreated kraft BL. The feed for the acid and the base compartment was in both cases 0.05 M NaOH solution. The feed for the electrode rinse solution was 1 M NaOH.

The experiment was performed at a constant electric potential of 10 V, the current was tracked over time. The experiment was performed for 9 h, samples were taken regularly from the acid, the base and the salt compartment. Due to the formation and accumulation of H^+^ in the acid compartment, the pH dropped over the course of the experiment. To yield the free carboxylic acids as a product, the pH was not adjusted.

#### 2.3.2. EDBM Treatment of BL Model Solution

The composition of the salt model solution was adjusted to the composition of the BL permeate in terms of formic, acetic, glycolic and lactic acid concentration as well as pH value. The pH value was adjusted with 50 g/L of anhydrous Na_2_CO_3_ and with NaOH pellets to the final pH of 10.4. The compositions of the feed solutions in the acid, base and the electrode rinse compartments were identical to that used in the ED treatment of BL permeate.

The ED experiment with BL model solution was performed at constant electric potential difference of 10 V for 25 h. Samples of the acid, base and salt compartment were taken repeatedly and the current was tracked over the whole experiment. The pH was checked regularly with pH indicator paper. To prevent the protonation and subsequent diffusion of protonated acids through the ion exchange membranes, the solution in the acid compartment was alkalized to assure a pH between 9 and 12.5. As soon as it dropped to pH 9, NaOH pellets (5–7 g) were dissolved in 10 mL of solution withdrawn from the acid compartment, and mixed back into the bulk liquid. In total, 20.7 g/L of NaOH was added to the acid compartment.

### 2.4. Analyses

#### 2.4.1. DM, Ash and pH Value

For analysis of the total DM content in the samples, an aliquot was dried to constant mass in a crucible at 105 °C. From the dried sample, the ash content was determined by ashing the sample at 600 °C to constant mass, typically for 24 h. All measurements were performed in triplicate.

For determination of the pH value of the liquid samples, an SI Analytics A164 1M-DIN-ID pH electrode (Xylem, Rye Brook, NY, USA) was used, and measurements were performed at room temperature.

#### 2.4.2. Lignin Content

The lignin content in the samples was estimated with UV-spectroscopy. The samples were diluted in 0.1 M NaOH before the absorbance was measured at a wavelength of 280 nm. The transmitted path length was 1 cm, and the absorption coefficient was 23.1 L/(g·cm) as determined by calibration with isolated kraft lignin. Measurements were performed using a Shimadzu UV-1800 Spectrophotometer (Shimadzu, Kyoto, Japan). All measurements were performed in triplicate.

#### 2.4.3. Carboxylic Acid Content

The concentrations of the low-molecular-weight carboxylic acids acetic acid, formic acid, glycolic acid and lactic acid were determined by HPLC measurements. All samples were acidified with H_2_SO_4_ to a pH of 1–2 to precipitate lignin and assure constant conditions. The acidified samples were diluted with deionized water and filtered through a 0.45 µm syringe filter. To each sample, 100 mg/L of DMSO was added as internal standard. Measurements were performed with a Dionex Ultimate 3000 HPLC system (Thermo Fisher Scientific, Waltham, MA, USA) equipped with a REZEX-ROA column (Rezex™ ROA-Organic Acid H+ 8%, LC Column 300 × 7.8 mm, Ea from Phenomenex). The detection of peaks was based on the UV absorption at 210 nm. Elution was achieved with 0.5 mL/min of 0.0025 M H_2_SO_4_ at 30 °C. The software Chromeleon 7 (Thermo Scientific, Waltham, MA, USA) was used to interpret the chromatograms. All measurements were performed in triplicate.

### 2.5. Evaluation of EDBM Experiments

To evaluate the EDBM process performance, the current efficiency and the electric energy consumption for carboxylic acid separation were calculated [26]. The current efficiency η [%] for the recovery of acids in the acid compartment was calculated according to Equation (1).
(1)$$\eta =\frac{z\cdot F\cdot n_{Acids,t}}{N\cdot \sum_{t=0}^{t}\left(I_{\Delta t}\cdot \Delta t\right)}\cdot 100$$

With z being the ion valence of the carboxylic acids (in this case z = 1), F being the Faraday constant (F = 96,485 C/mol), $n_{Acids,t}$ being the moles of acids transported from the salt to the acid compartment at the time t (in mol), N being the number of cell triplets in the EDBM stack (in this case N = 5), $I_{\Delta t}$ being the average current during a time interval (in A), and Δt being the respective time interval (in s).

The electric energy requirement E (in kWh/kg_Acid_) for transport of acids to the acid compartment was calculated according to Equation (2).
(2)$$E=\frac{U\cdot \sum_{t=0}^{t}\left(I_{\Delta t}\cdot \Delta t\right)}{m_{Acids,t}\cdot 1000}$$

With U being the electric potential difference (here U = 10 V) and $m_{Acids,t}$ being the total mass of acids transported from the salt compartment to the acid compartment at time t (in kg).

## 3. Results and Discussion

### 3.1. BL Feed Pretreatment by NF

NF of the pretreated BL was conducted to separate lignin from the feed stream. The negatively charged, large-molecular-weight lignin molecules could otherwise lead to concentration polarization effects and serious fouling of the ED membranes, as was already reported by other authors [19,20]. At least 2 L of treated BL feed solution is required to operate the ED system, thus several batches of NF were necessary to yield the required quantity of permeate. The NF treatment was conducted until a volume reduction of up to 50% was reached as higher volume reductions led to a significantly reduced permeate flow rate. Exemplarily, a typical development of the flow rate with volume reduction is shown in Appendix A. Table 2 summarizes the changes in concentration of the BL constituents in the NF product streams for a representative NF experiment.

By NF, the lignin concentration was reduced below 30% of the original value. For the considered range of volume reduction, 80–90% of the total lignin from the feed stream was recovered in the retentate. Due to enrichment of lignin and other dissolved BL constituents, the DM content of the retentate increased to ~120% of the feed’s content and also ash was enriched to ~110%.

With respect to the evaporation train in a mill, reintroduction of this retentate would be beneficial in terms of necessary evaporator capacity. Alternatively, the lignin could be isolated from this enriched stream, e.g., by acid precipitation, and be further processed to value added bio-based products. The membrane fractionation typically narrows the molecular mass distribution compared to the original kraft lignin fraction in the untreated black liquor [27], which may be advantageous for applications where defined properties of lignin are required. In contrast to the lignin, the pulping chemicals need to be recycled to the pulping process as much as possible. The permeate stream contains a significant fraction of the ash and thus the inorganic pulping chemicals. The recovery of especially NaOH from BL permeate could be realized in a subsequent EDBM treatment.

Together with NaOH, the carboxylic acids should be recovered from the BL permeate by ED. As shown in Table 2, the pretreatment with NF already led to an increase in acid concentration in the permeate compared to the feed. This treatment thus not only acted as a pre-purification but also as a pre-concentration step. The increase in acid concentration in the permeate was, at 23%, significant and similar for all four investigated acids. Overall, a typical NF experiment led to an acid recovery of 55–65% in the permeate. The reason for the higher acid concentration in the permeate compared to the feed/retentate of a NF experiment is the Donnan equilibrium. At the high liquor pH value, the phenolic groups of the dissolved lignin in BL were negatively charged. As the majority of lignin could not pass the membrane, the negatively charged carboxylic acid anions were crossing the membrane faster than the solvent to maintain the electro-neutrality between the permeate and the retentate solutions. Similar findings were reported by Mänttäri et al. [16].

In summary, the NF pretreatment led to a sufficient reduction of lignin concentration and DM content and an increase in carboxylic acid concentration in the permeate. Both changes are beneficial for the subsequent ED treatment. The pooled permeate that was used as feed for ED had a DM content of 16%, of which around 5% was residual lignin, corresponding to a total lignin concentration of approx. 8 g/kg. Due to the NF treatment, it is assumed that this residual lignin was of low-molecular-mass and had a high solubility in water at neutral or alkaline pH. Further, around 30% of the DM in the permeate consisted of the four measured carboxylic acids.

### 3.2. EDBM Treatment of BL Permeate

The EDBM experiment with BL permeate was conducted for 9 h at a constant cell voltage of 10 V, and the current was monitored throughout the experiment. The resulting development of power is plotted in Figure 2, showing a constant decrease in power consumption during the course of the experiment. The current decreased from 0.69 A at the beginning to 0.33 A at the end of the experiment, corresponding to current densities between 69 and 33 A/m^2^.

An evaluation of the lignin concentration in the three product compartments after the experiment and a visual evaluation of the samples proved that no lignin or chromophoric compound had migrated into the acid or the base compartment. The comparably larger and slower lignin molecules were completely retained by the applied anion exchange membrane despite the negative lignin charge. The development of the carboxylic acids concentration in the three compartments compared to the feed BL permeate is plotted in Figure 3a–c. The concentration of acids in the salt compartment (Figure 3a) decreased throughout the experiment for all four acids. Correspondingly, the acid concentrations increased in the acid product compartment (Figure 3b). There was a clear correlation between the rate of concentration change and the size and thus diffusivity of the investigated acids. Formic acid, the smallest compound, migrated the fastest from the salt to the acid compartment, while the largest acid, lactic acid, migrated the slowest in the electric field. This is in agreement with published data, where a correlation between migration velocity and molecular size during ED experiments is also reported [28,29]. The high diffusivity of formic acid led to a significant loss of acid to the base compartment, as can be seen from the development of acid concentration shown in Figure 3c. After approx. 3 h of treatment, the concentration increase in formic acid in the acid compartment started to level off and reached a final value of 45% of the formic acid concentration in the feed. At the same time, the concentration of formic acid in the base compartment started to increase linearly, reaching a final value of 30% of the formic acid concentration in the feed, with a rising trend until the end of the experiment. In combination with the decrease in formic acid content in the salt compartment, these developments show that towards the end of the experiment, the migration rate of formic acid from the salt to the acid compartment was nearly the same as the diffusion velocity from the acid to the base compartment through the BPM. The reason for that is again the high diffusivity of formic acid [20,21]. Approximately 3 and 5% of acetic and glycolic acid, respectively, also ended up in the base compartment, while almost no lactic acid was detected. The undesired diffusion of acids to the base compartment was strongly dependent on the pH in the acid compartment. A significant diffusion only happened as the pH dropped to the range of the acid’s pK_a_ values (between 4.76 for acetic acid and 3.75 for formic acid [30]) due to the formation of H^+^ in the BPM. The carboxylic acids that had accumulated in the acid product compartment were then partially protonated and thus not retained by the BPM and AEM anymore and could diffuse back into the salt compartment, reducing the current efficiency. Further, they diffused through the BPM to the base compartment, where they were deprotonated by the high OH^−^ concentration and thus accumulated in the base compartment. Overall, this effect led to a loss of 8% of the acids, mainly formic acid, to the base compartment. In the acid product compartment, 17% of the total acids were found. A mass balance of the individual acids yielded a balance error between 6 and 9%, which can likely be attributed to the evaporation of water and transfer of water by electroosmosis caused by the transfer of hydrated ions to the acid and base compartments. In the used set-up, the change of total mass or volume in the compartments could not be monitored throughout the experiment, and significant dead volumes only allowed for a rough estimation of the final volumes. Therefore, the evaporation and migration of water throughout the experiment could not be determined. In consequence, all results are given based on constant volumes in all three compartments.

Figure 3d shows the initial and final DM and ash content in all compartments. Very little ash was found in the acid compartment, indicating that the migration of sulphate from the salt to the acid compartment was negligible. Potential migration of hydrogen sulphide and carbonates cannot be assessed by ash content measurements, however, as the low pH leads to formation of volatile H_2_S and CO_2_. The ash content of the final base compartment with approx. 1.3% indicates a significant migration of inorganic ions, presumably Na^+^, to the base compartment, forming NaOH which could be recycled directly to the kraft recovery cycle.

The near-linear development of carboxylic acid concentrations in the salt and the product compartment implies that no build-up of a lignin fouling layer on the membrane surface took place. Additionally, no visible fouling layer was found on the membranes upon optical inspection. Hence, the lignin removal by NF treatment was sufficient and shows that the application of a combined membrane treatment sequence consisting of NF and ED is a promising option for fractionation of kraft BL to simultaneously isolate lignin and carboxylic acids. However, while the DM and ash measurements showed that the acid product was fairly pure, the acid recovery of 17% needs to be significantly improved and the loss of acids to the base compartment controlled. To assess the influence of pH in the acid compartment and to evaluate longer treatment times on acid yield, a simplified EDBM long-term experiment with a BL model solution was carried out.

### 3.3. EDBM Treatment of BL Model Solution

The BL model solution contained the four carboxylic acids formic, acetic, glycolic and lactic acid in a similar ratio as the BL permeate. The pH was adjusted to 10.4 by addition of Na_2_CO_3_ and NaOH. The overall carboxylic acid concentration in the first sample of the salt compartment was approx. 20% lower in the model solution than in the BL permeate. This deviation was caused by the high dead volume in the compartment, which still contained water from cleaning the system.

Throughout the experiment, which lasted 25 h, the pH in the acid compartment was kept above pH 9 by regular addition of NaOH, with the goal to prevent protonation of the accumulated carboxylic acids and to limit loss to the base compartment. Figure 4 shows the recorded power over treatment time. The red circles indicate the addition of NaOH. The addition of alkali to the acid compartment significantly increased the current after each addition. This effect was especially dominant during the first 4 h of the experiment and indicates that during this period the electrical resistance in the acid compartment had a dominant influence on the resistance of the whole stack. Correspondingly, conductivity measurements of the products of the EDBM experiment with BL permeate showed that the conductivity in the acid compartment was only in the range of 10% of that in the base and the salt compartment (see Appendix A). The resulting higher electrical resistance was one reason for the lower current density in the experiment with BL permeate.

Figure 5a–c show the development of the carboxylic acid concentrations and pH values in the three EDBM compartments. The general trends between the BL permeate and the BL model solution were similar. The migration velocity of formic acid was the highest, and after 13 h the salt compartment (Figure 5a) was completely depleted of formic acid. Towards the end of the experiment glycolic acid and acetic acid were also depleted from the salt stream, up to a final concentration below 2% of the initial content. Only the concentration of lactic acid was still 27% of the initial value, although it was steadily decreasing. Correspondingly, the acid concentrations in the acid compartment (Figure 5b) increased. Although the control of pH value significantly limited the loss of acids to the base compartment (Figure 5c), diffusion was still an important factor. After 9 h of treatment, 2% of the total acids were found in the base compartment, compared to 8% during the experiment without pH control and BL permeate. However, after 25 h, 12% of the total acids had migrated to the base compartment. This was also reflected in a reduction of the acid concentration in the acid compartment towards the end of the experiment. The trends clearly show that the unwanted migration of acids took place from the acid to the base compartment and not from the feed compartment. Overall, 72% of the total acids were found in the acid compartment. The yields were again calculated under the simplification that the volumes in all compartments remained constant throughout the experiment. This could not be completely achieved during this experiment. Evaporation and migration of water led to a visible reduction of volume in the salt compartment, while only a little increase in the volume of the acid and no increase in volume in the base compartment could be determined.

The mass balance yielded errors up to 6%, except for formic acid, where seemingly 38% of the initially added acid was lost. As small amounts of formic acid were also detected in the electrode rinse solution, a likely reason for that discrepancy is that formic acid was oxidized at the electrodes to form CO_2_ [31].

The results of the DM and ash measurements shown in Figure 5d underline the potential to also remove Na^+^ from the salt solution. The ash content in the salt compartment was reduced from initially 4.1% to a final concentration of 1.2%. Correspondingly, the ash content in the base compartment increased to 2.8%. The final ash content in the acid compartment fits, at 2.4%, closely to the amount of NaOH that was added throughout the experiment.

### 3.4. Evaluation of EDBM Experiments

The two EDBM experiments were evaluated based on current efficiency, power consumption and hourly yield of carboxylic acids. The calculations were based on the measured yield in the acid compartment only. The development of current efficiency for both experiments is depicted in Figure 6. The initial current efficiency in both experiments was similar, but the decrease was faster for the experiment with BL permeate. A reason for the decrease in current efficiency is the loss of acids by diffusion back into the feed and the base compartment, as well as the competing migration of OH^−^ and potentially other anions. Comparable current efficiencies were reported by Kumar and Alén [14] for recovery of sodium ions from dilute, carbonated soda BL.

As the current efficiency decreased, the power consumption for the recovery of carboxylic acids increased over the course of the experiments in both cases. The results are shown in Table 3, indicating again that power consumption increased more rapidly for the experiment with BL permeate.

Finally, the hourly yield of acids was compared for both experiments; the results are shown in Figure 7. In agreement with the development of current density and current efficiency, the acid recovery in the acid compartment was generally faster in the experiment with BL model solution. This shows the need for further improvement of the process. From the results of the model solution experiment it can be shown that maintaining a high pH value in the acid compartment considerably increases the recovery as it slows diffusion of acids to the base compartment and increases the electrical conductivity in the acid compartment. This in turn leads to a higher current efficiency and acid recovery rate and can, together with a longer treatment time, also lead to an increase in acid recovery upon treatment of BL permeate. An increase in the available membrane area could also improve the acid yield as it decreases the necessary processing time and thus limits the available time for unwanted diffusion of acids to the base compartment. Industrial-scale EDBM stacks can comprise more than 100 unit cells with much higher membrane areas (e.g., 1 m^2^) compared to laboratory-scale equipment [32], showing the considerable upscaling potential.

However, a disadvantage of a process with the addition of NaOH to the acid compartment is the generation of carboxylates instead of the free acids. Another option is the use of EDBM with alternating bipolar and anion exchange membranes (anion stripping ED), as was applied by Wang et al. to purify carboxylic acids [33]. Such a set-up would generate a purified acid stream and an alkaline stream of sodium hydroxide, lignin and neutral BL constituents that could be recycled to the recovery process. As a disadvantage, however, no pure NaOH stream is generated.

## 4. Conclusions

The goal of the present study was to fractionate a kraft BL stream to recover low-molecular-weight carboxylic acids and NaOH. The BL was first pre-purified by NF. The concentration of lignin in the resulting permeate was reduced to below 30%, and the concentration of carboxylic acids increased to more than 120% compared to the BL feed. This permeate was treated by EDBM to isolate the carboxylic acids and recover NaOH. After 9 h of treatment, 17% of the acids were recovered in the acid compartment, while 8% diffused to the base compartment. No migration of lignin was observed. A comparative experiment with BL model solution showed that pH control can limit the loss of acids to the base compartment. The experiment with model solution yielded higher current efficiencies and acid recovery rates than that with BL permeate. In this experiment, 68% of the ash from the salt compartment was recovered in the base compartment, and 72% of the carboxylic acids were recovered in the acid compartment. These results underline the high potential of co-recovery of carboxylic acids and NaOH using EDBM.

## Figures and Tables

**Figure 1 membranes-13-00092-f001:**
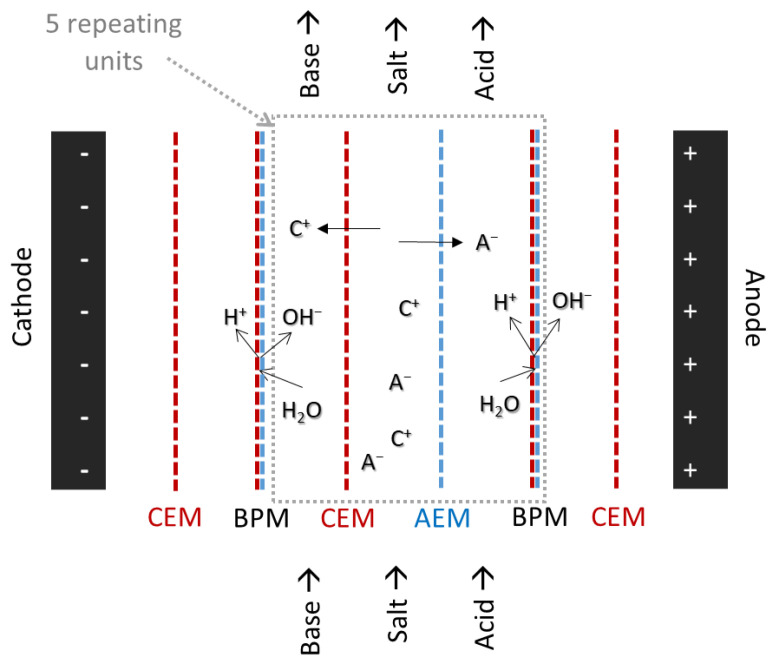
Schematic of membrane configuration and separation mode in the EDBM membrane stack.

**Figure 2 membranes-13-00092-f002:**
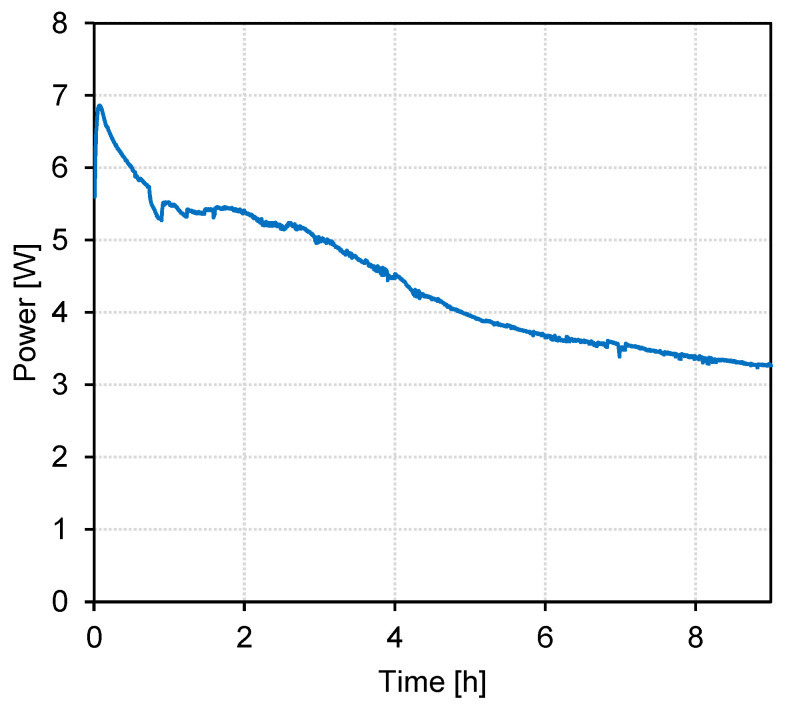
Power over treatment time for the batch EDBM treatment of BL permeate, U_cell_ = 10 V (constant).

**Figure 3 membranes-13-00092-f003:**
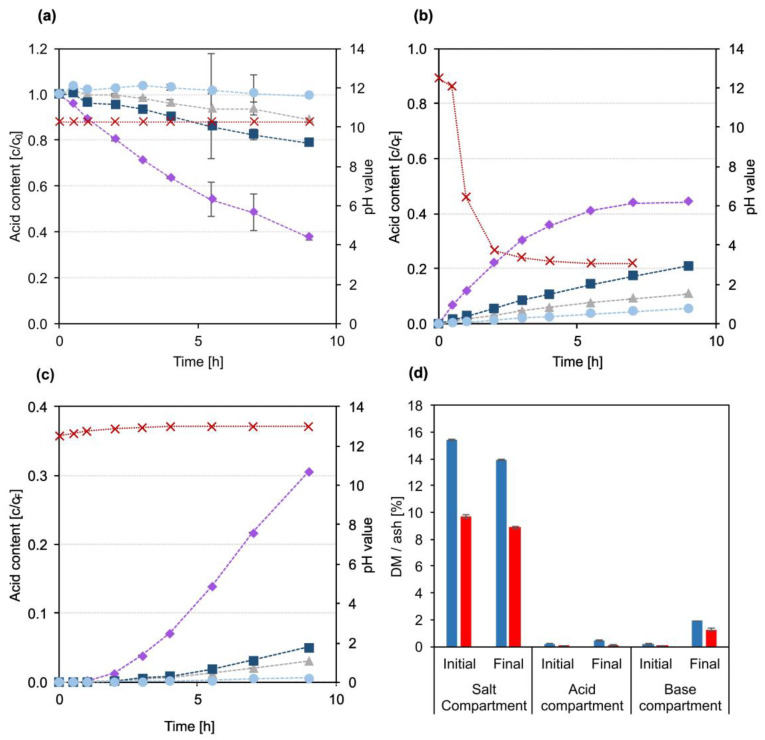
Development of carboxylic acid concentrations and pH value over the course of the EDBM experiment with BL permeate. (**a**) Salt compartment; (**b**) acid compartment; (**c**) base compartment. For (**a**–**c**): ■ glycolic acid, ● lactic acid, ♦ formic acid, ▲ acetic acid, **X** pH value. (**d**) DM (■) and ash (■) content measured in the initial and final samples from the three compartments.

**Figure 4 membranes-13-00092-f004:**
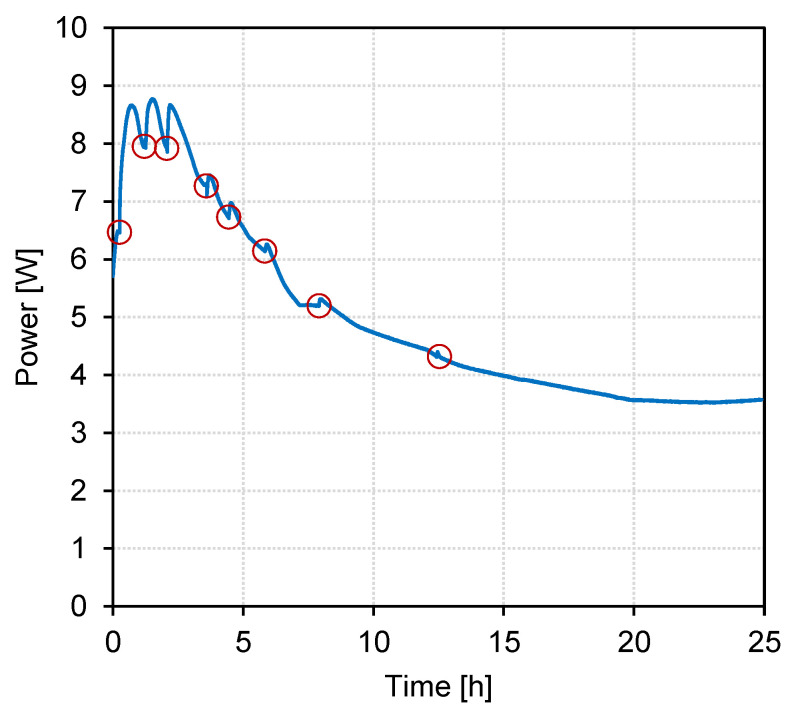
Power over treatment time for the batch EDBM treatment of BL model solution, U_cell_ = 10 V (constant). The red circles mark the points of NaOH addition.

**Figure 5 membranes-13-00092-f005:**
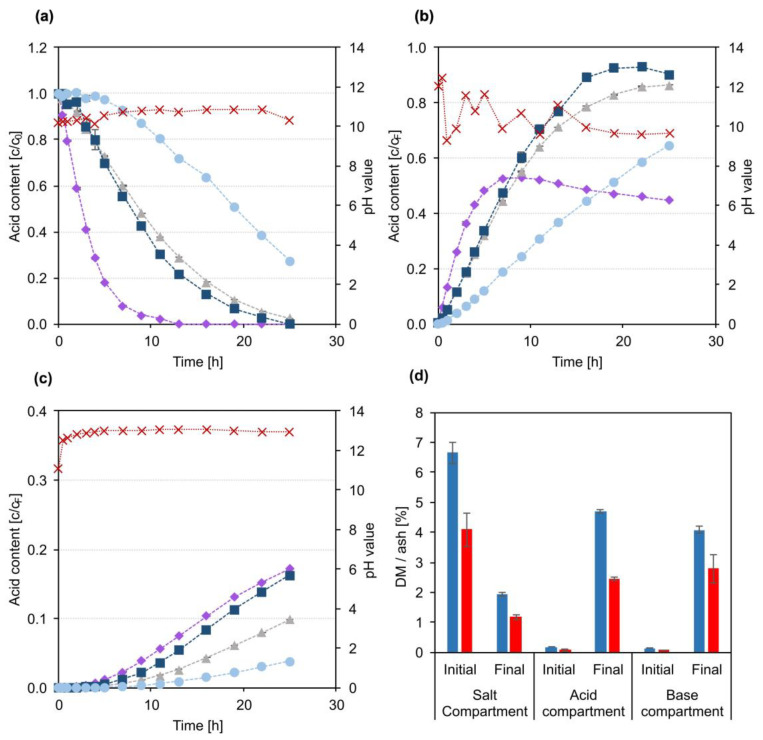
Development of carboxylic acid concentrations and pH value over the course of the long-term EDBM experiment with BL model solution. (**a**) Salt compartment; (**b**) acid compartment; (**c**) base compartment. For (**a**–**c**): ■ glycolic acid, ● lactic acid, ♦ formic acid, ▲ acetic acid, **X** pH value. (**d**) DM (■) and ash (■) content measured in the initial and final samples from the three compartments.

**Figure 6 membranes-13-00092-f006:**
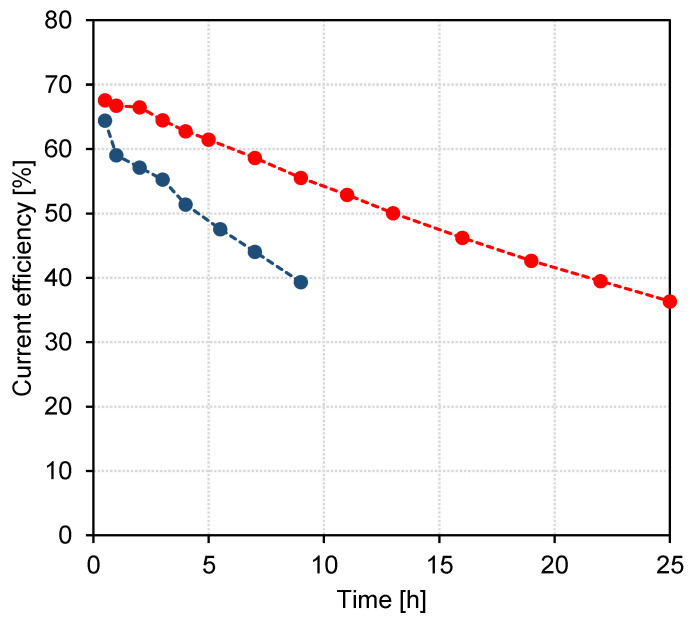
Current efficiency development during the EDBM experiment with BL permeate (●) and BL model solution (●).

**Figure 7 membranes-13-00092-f007:**
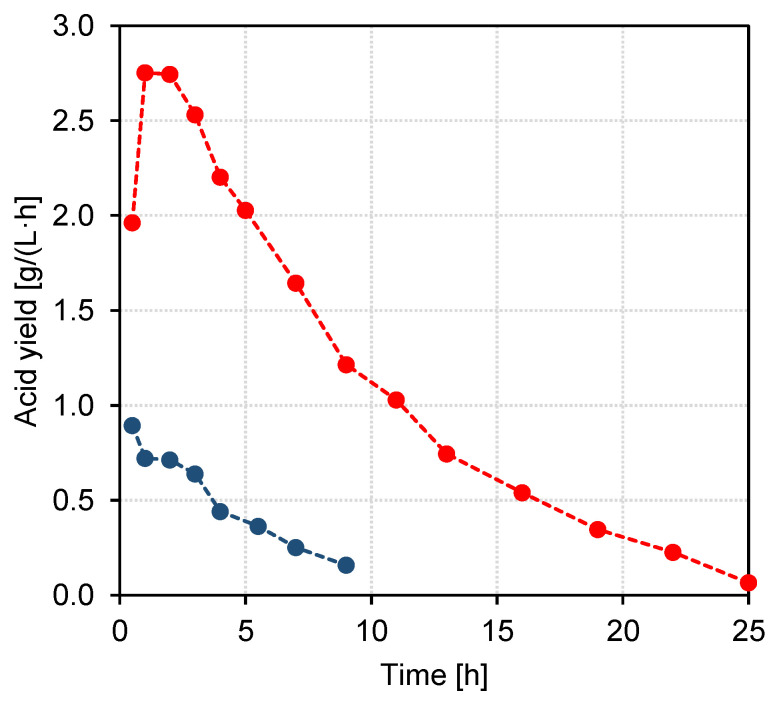
Yields of total acids in the acid compartment for the EDBM experiment with BL permeate (●) and BL model solution (●).

**Table 1 membranes-13-00092-t001:** Composition and characteristics of the pretreated BL feed.

Parameter	
Wood source	Hardwood
pH	12.1
Dry matter (DM) [%]	29.1
Sum measured carboxylic acids [g/kg]	65.6
Formic acid [g/kg]	19.9
Acetic acid [g/kg]	34.0
Glycolic acid [g/kg]	2.4
Lactic acid [g/kg]	9.3
lignin [g/kg]	45

**Table 2 membranes-13-00092-t002:** Relative concentrations of the BL constituents in the permeate (c_P_) and retentate (c_R_) stream of a typical NF experiment. Data for NF treatment of pretreated BL to a volume reduction of 37%.

	Permeate c_P_/c_Feed_	Retentate c_R_/c_Feed_
Lignin	0.28	1.41
Carboxylic Acids	1.23	0.87
DM	0.78	1.18
Ash	0.88	1.09

**Table 3 membranes-13-00092-t003:** Electrical energy requirement for the EDBM experiments with BL permeate and BL model solution after different treatment times for recovery of total carboxylic acids.

	EDBM Treatment Time [h]	Electrical Energy Requirement [kWh/kg_Acid_]
BL permeate	1	1.8
	9	2.6
BL model solution	1	1.5
	9	1.7
	25	2.5

## Data Availability

Not applicable.

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
