# Peer review of "Isolation of Carboxylic Acids and NaOH from Kraft Black Liquor with a Membrane-Based Process Sequence"

_membranes, 2023, doi:10.3390/membranes13010092_

Round 1

Reviewer 1 Report

It is an interesting manuscript with valuable findings within fractionation of black liquor. There are some iterations that would elevate the manuscript and some curiosities to be addressed. 

  The fractionation is demonstrated for the BL permeate that demonstrates a modest acid yield. The process is then improved by pH control for a BL model system containing the target acids. Would be beneficial if the authors can comment on what are the possibilities and limitations regarding iterating the BL permeated retentate with the learnings of the model solution. That is, how to make the system work for BL.   In the Figure 3 caption it may be misleading that the ash and DM content is reported “of the compartments”. Would it not be actuaaly of the fractions in these compartments?   It would increase comparability of the Figures 3 and 4 had the same scale (3b,b,d and 4 b,c, d) and accuracy (4c). Looking at the c-figures then reveales that in the case of the BL permeate, formic acid is challenging the experiment, yet in the model system glycolic acid content in the base compartment is higher than in the BL permeate fractionation. The reasons for this needs to be clarified.    The abstract reports yield of 69 shile conclusions states 68 %. Please revise.    There are some readibility issues in the chapter starting line 234. This needs a rewrite.

Author Response

  • The fractionation is demonstrated for the BL permeate that demonstrates a modest acid yield. The process is then improved by pH control for a BL model system containing the target acids. Would be beneficial if the authors can comment on what are the possibilities and limitations regarding iterating the BL permeated retentate with the learnings of the model solution. That is, how to make the system work for BL.

Thank you for the comment, we tried to provide these implications already in section 3.4, in the revised version of the manuscript we elaborated further on the subject according to this comment, by implementing a section (line 468-476).

  • In the Figure 3 caption it may be misleading that the ash and DM content is reported “of the compartments”. Would it not be actually of the fractions in these compartments?

Thank you for this remark, the respective sections in the figure 3 and figure 5 captions were rewritten.

  • It would increase comparability of the Figures 3 and 4 had the same scale (3b,b,d and 4 b,c, d) and accuracy (4c). Looking at the c-figures then reveales that in the case of the BL permeate, formic acid is challenging the experiment, yet in the model system glycolic acid content in the base compartment is higher than in the BL permeate fractionation. The reasons for this needs to be clarified.

We adjusted the scales of the respective y-axis so that they are equal among the sub-figures in Figure 3 and Figure 5, respectively, for all figures except 3d/5d, as we feel that too much information would be lost if Figure 5d was rescaled.

Regarding the seemingly different behavior of glycolic acid, it needs to be pointed out that the glycolic acid fraction in the base compartment after 9 hours of treatment of BL model solution is still by a factor of 2 lower than after the same time of treatment of BL permeate. Here it is important to point out that the glycolic acid concentration in all the samples was considerably lower than that of the other acids (see Table 1) which makes it much more prone to measurement errors. Furthermore, the difference in current density for the two experiments affects the ratio of electromigration to diffusion differently for all four acids, which can also be partially responsible for the differences observed for the two experiments. Unfortunately, these discussions are very speculative and thus we would prefer to not include them in the manuscript.

  • The abstract reports yield of 69 shile conclusions states 68 %. Please revise.

Thank you very much for pointing this out, we revised and corrected the numbers accordingly.

  • There are some readibility issues in the chapter starting line 234. This needs a rewrite.

We assume this comment is related to the last comment from Reviewer 3, who states that the caption of Table 2 is included in this text fragment. This is, however, not the case for the version of the manuscript that we, as authors, find on the online system. To remove the risk of a cross-reference like this in the final manuscript, we re-inserted and adjusted the whole paragraph.

Reviewer 2 Report

Dear editor,

Thank you for the invitation to review the work by Kienberger and co-workers on the fractionation of kraft black liquor using a membrane sequence.

Pre-treated should be pretreated (and pre-treatment pretreatment) throughout the manuscript

In table 1: wood source is indicated to be hardwood. Is there a further specification possible? If you receive a sample from the mill, then most likely at the mill they know what was the main wood source for that sample.

Author Response

  • Pre-treated should be pretreated (and pre-treatment pretreatment) throughout the manuscript

We changed the words according to the reviewers suggestion.

  • In table 1: wood source is indicated to be hardwood. Is there a further specification possible? If you receive a sample from the mill, then most likely at the mill they know what was the main wood source for that sample.

Thank you for your comment, we now included the specifications we received from the mill (lines 94-95).

Reviewer 3 Report

The manuscript “Isolation of carboxylic acids and NaOH from kraft black liquor with a membrane based process sequence” reports the application of nanofiltration to separate the lignin from kraft black liquor prior to the separation of carboxylic acids and inorganics by electrodialysis with bipolar membranes. As the authors describe in the introduction, the isolation of lignin from black liquors is already an industrial reality, however the isolation of carboxylic acids is not done in large scale mainly due to the heterogeneous acid composition and lack of economically feasible separation protocols. These carboxylic acids are present in high concentrations (~30%) in the black liquor and their contribution to energy production in the mills is only minor, therefore the development of separation processes for their isolation and further utilization in the manufacture of bio-based products is of relevance.

The manuscript is well written and relatively easy to understand, the results are presented in a clear and logical manner and the conclusions are supported by the results obtained. I recommend the publication of the manuscript in the journal Membranes after the authors have addressed the following comments:

·       The possible uses of carboxylic acids in the development of materials and chemicals, with references to published literature, should be mentioned in the introduction to justify the need for their recovery.

·      What is the current state-of-art with respect to utilization of membrane processes, such as nanofiltration and electrodialysis, in industrial environments? This should be indicated in the introduction for the benefit of the reader.

·       Is the process developed by the authors potentially scalable? This should be briefly mentioned in the discussion of the results or conclusions.

·       Line 90: “The pre-treated kraft BL was supplied by a local pulp mill”. What sort of pre-treatment was done to the black liquor before delivery and utilization in the current study?

·       Lines 228-229:  “The presumably narrowed molecular mass distribution compared to kraft lignin in untreated black liquor may be advantageous for applications where defined properties of lignin are required”. Is this statement based on experimental data? There is no data on molar mass distribution shown in this study. If the authors refer to data obtained in previous studies, please add the corresponding reference(s).

·       Lines 236-238: The whole caption of Table 2 is included in the text. Please adjust the cross-reference function.

Author Response

  • The possible uses of carboxylic acids in the development of materials and chemicals, with references to published literature, should be mentioned in the introduction to justify the need for their recovery.
  • What is the current state-of-art with respect to utilization of membrane processes, such as nanofiltration and electrodialysis, in industrial environments? This should be indicated in the introduction for the benefit of the reader.

Thank you for this comment, the introduction was adapted according to these suggestions. The additions can be found in lines 44-45 and 61-63 of the revised manuscript.

  • Is the process developed by the authors potentially scalable? This should be briefly mentioned in the discussion of the results or conclusions.

A comment regarding upscaling was added to the discussion section, lines 474-477.

  • Line 90: “The pre-treated kraft BL was supplied by a local pulp mill”. What sort of pre-treatment was done to the black liquor before delivery and utilization in the current study?

We received the BL already in the pre-treated form, unfortunately the pre-treatment performed by the mill is confidential. But, as can be seen from the composition summarized in Table 1, the black liquor we received is comparable to black liquors reported in literature, which are already quite diverse by nature, depending on the wood source, cooking conditions, etc.

  • Lines 228-229:  “The presumably narrowed molecular mass distribution compared to kraft lignin in untreated black liquor may be advantageous for applications where defined properties of lignin are required”. Is this statement based on experimental data? There is no data on molar mass distribution shown in this study. If the authors refer to data obtained in previous studies, please add the corresponding reference(s).

This statement is based on findings by other authors, we did not determine the molecular mass in the generated samples. We adjusted the respective section to avoid misunderstandings and included a literature reference regarding lignin fractionation with membrane processes.

  • Lines 236-238: The whole caption of Table 2 is included in the text. Please adjust the cross-reference function.

Thank you for this comment, this obviously was an mistake by the software, when uploading the manuscript. To remove the risk of a cross-reference like this in the final manuscript, we re-inserted and slightly adjusted the whole paragraph.